

# Nuclear spin noise tomography in three dimensions

Stephan J. Ginthör[1], Judith Schlagnitweit[1,2], Matthias Bechmann[1], Norbert Müller[1,3*]

[1]Institute of Organic Chemistry, Johannes Kepler University Linz, Linz, 4040, Austria
[2]Current affiliation: Department of Medical Biochemistry and Biophysics, Karolinska Institutet, Stockholm, SE-171 77,
Sweden
[3]Faculty of Science, University of South Bohemia, České Budějovice, 37005, Czech Republic

*Correspondence to*: Norbert Müller (norbert.mueller@jku.at)

**Abstract.** We report three-dimensional spin noise imaging (SNI) of nuclear spin density from spin noise data acquired by
Faraday detection. Our approach substantially extends and improves the two-dimensional SNI method for excitation-less
magnetic resonance tomography reported earlier. (Müller, N. and Jerschow, A.: Nuclear spin noise imaging, Proc. Natl. Acad.
Sci. U.S.A., 103(18), 6790–6792, doi:10.1073/pnas.0601743103, 2006.) This proof of principle was achieved by taking
advantage of the particular continuous nature of spin noise acquired in the presence of constant magnitude magnetic field
gradients and recent advances in nuclear spin noise spectroscopy acquisition as well as novel processing techniques. In this
type of projection-reconstruction based spin noise imaging the trade-off between signal-to-noise ratio (or image contrast) and
resolution can be adjusted a posteriori during processing of the original time domain data by iterative image reconstruction in
a unique way not possible in conventional rf-pulse dependent MRI. The 3D SNI is demonstrated as a proof of concept on a
commercial 700 MHz high resolution NMR spectrometer, using a 3D-printed polymeric phantom immersed in water.

## 1 Introduction

The phenomenon of nuclear spin noise, first predicted by Felix Bloch in 1946 (Bloch, 1946), can be ascribed to the incomplete
cancellation of the fluctuating spin magnetic moments in a specimen. Owed to the extremely low amplitudes of these residual
fluctuations of the bulk magnetic moment very low-noise rf (radio-frequency) circuitry is required to separate nuclear spin
noise signals from background noise (Müller et al., 2013) (Ferrand et al., 2015) (Pöschko et al., 2017). Therefore, experimental
detection of nuclear spin noise succeeded only in 1985 (Sleator et al., 1985). Today readily available low-noise rf-electronics
enable one to observe nuclear spin noise in reasonable amounts of time, in particular if a cryogenically-cooled probe circuit is
used (Kovacs et al., 2005) (Müller and Jerschow, 2006). In spite of low intrinsic sensitivity, nuclear spin noise based
spectroscopy and imaging techniques have an intriguing potential, mainly for three reasons: (1) The spin noise signal
magnitude exceeds the thermal polarization derived signal for very low numbers ($<\sim 10^8$) of nuclear spins as it scales with the
square root of the number of spins and not linearly. (2) The observation of undisturbed spin systems becomes possible. (3)
Signals acquired in absence of rf-pulses are devoid of limitations imposed by pulse imperfections and bandwidth. The
theoretical and technical aspects of nuclear spin noise detected by Faraday induction have been studied extensively in recent





years (Marion and Desvaux, 2008) (Nausner et al., 2009) (Desvaux et al., 2009) (Müller et al., 2013) (Chandra et al., 2013) (Ferrand et al., 2015) (Pöschko et al., 2017) (Ginthör et al., 2018). In the research we report here, a major sensitivity and image quality improvement in the case of spin noise imaging (SNI) (Müller and Jerschow, 2006) is achieved by exploiting the tuning dependence of the spin noise line shape (Pöschko et al., 2014).

## 1.1 Magnetic resonance imaging

Spatial resolution in magnetic resonance imaging (MRI) relies on frequency encoding the positions in magnetic field gradients (Kumar et al., 1975). Thus, spectra recorded in presence of magnetic field gradients can be interpreted as the result of a forward Radon transformation applied to the spin density function of the sample along the gradient direction (Deans, 2007).

Most of today's MRI approaches are based on the idea of Fourier imaging originally introduced by Richard R. Ernst (Kumar et al., 1975) and use rf pulses preceding excitation and evolution periods, in which phase encoding via field gradients is attained, followed by a two-dimensional (2D) Fourier transformation (FT) (Wright, 1997) (Brown et al., 2014). The resolution in the third dimension is commonly obtained using slice-selection, i.e. by applying rf-pulses simultaneously with magnetic field gradients (Garroway et al., 1974). A multi-dimensional Fourier transform approach could, in principle, also work for noise data. However, it would suffer from even lower sensitivity due to transverse relaxation, in particular if no refocusing rf pulses are applied, which would counteract the goal of imaging without rf pulses. It should be noted here, that when detecting noise from very low numbers of spins refocusing pulses are a viable option, in particular if indirect detection by optical means is used (Meriles et al., 2010).

## 1.2 Projection reconstruction

Alternatively, multi-dimensional MRI can be based on the principle of reconstruction from projected data in the direct (frequency or spatial) observation dimension (Chetih and Messali, 2015), which closely corresponds to the only approach available in radiation-based computer tomography (CT). In the case of MRI, projections are provided through acquisition of free induction decays while different magnetic field gradients spanning the entire directional space are applied sequentially.

An intrinsic problem of the inverse Radon transform is that it almost always produces artefacts if the projections contain substantial noise contributions or if the imaged distribution function is not smooth (Kabanikhin, 2008). In particular the latter is a relevant restriction in most practical applications, as hard edges are very common features (e.g. bones in the human body). Therefore, alternatives to the inverse Radon transform are required. The processing protocol presented here aims to minimize these artefacts while maintaining resolution limited by transverse relaxation and maximizing the spin-noise to random-noise contrast.

Different from spectroscopic applications of magnetic resonance, where the chemical shift is of main interest, for imaging purposes one often assumes that all spins inside the imaged object have indistinguishable chemical shifts. We are going to use that assumption here, being aware that methods to cope with imaging artefacts caused by non-uniform chemical shifts (e.g.





water and fat) within a specimen exist (Dietrich et al., 2008) but may not applicable straightforwardly in the case of spin noise detection.

## 2. Results and Discussion

In order to optimize the spin noise detection, we take advantage of the progress in nuclear spin noise spectroscopy that has been achieved since the introduction of 2D nuclear spin noise imaging (Müller and Jerschow, 2006). In order to obtain symmetrical line shapes and optimize receiver sensitivity the cryogenically cooled NMR-probe is tuned to the SNTO (spin noise tuning optimum) (Marion and Desvaux, 2008) (Nausner et al., 2009) (Pöschko et al., 2014). As residual static field gradients can lead to spectral artefacts under conditions where radiation damping occurs (Pöschko et al., 2017) careful

optimization of the basic magnetic field homogeneity (achieved by shimming) and using sufficiently strong gradients (exceeding the broadening caused by radiation damping) are prerequisites for obtaining accurate nuclear spin noise images. We demonstrate nuclear spin noise 3D tomography on the phantom shown and described in Fig. 1 using the acquisition and processing procedures outlined in Sect. 3.

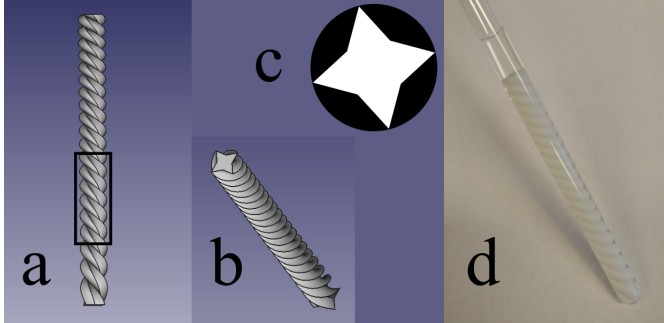

**Figure 1: Phantom used for the spin noise imaging experiments. (a) Rendered image of the phantom from a front perspective. (b) Rendered image of the phantom from a front/top perspective. (c) Typical cross section, which is a star-shaped octagon (white). This shape was rotated around and translated along the Z axis simultaneously to form the 3D phantom object. (d) The 3D printed phantom from polylactide (PLA) polymer inserted into a standard 5 mm NMR tube filled with $H_2O/D_2O$ (9:1). The black frame in (a) indicates the sensitive area within the NMR spectrometer's probe.**

The detected noise voltage was digitized quasi-continuously for each of 900 gradient directions uniformly distributed in 3D space. Each of the raw noise time domain data blocks was divided into overlapping windows, which were Fourier transformed and the resulting power spectral data added up to yield individual projections for each gradient setting. Our newly developed iterative projection-reconstruction protocol combines projections obtained by Fourier transform of the time domain noise data with different sliding window sizes (Desvaux et al., 2009) to obtain a 3D tomogram. This approach affords superior image

quality with respect to resolution and contrast as compared to using a single fixed sliding window size only. A final reconstructed image of the phantom obtained from the projections of spin noise with different gradients by the optimized iterative reconstruction based on the simultaneous algebraic reconstruction technique (SART) introduced by Andersen and Kak (Andersen and Kak, 1984) can be seen in Fig. 2a and d.

**MAGNETIC RESONANCE**
Open Access Discussions

In Fig. 2b and e we show the corresponding views, obtained using a fixed sliding window size of 1024 data points

(corresponding to 103 ms).

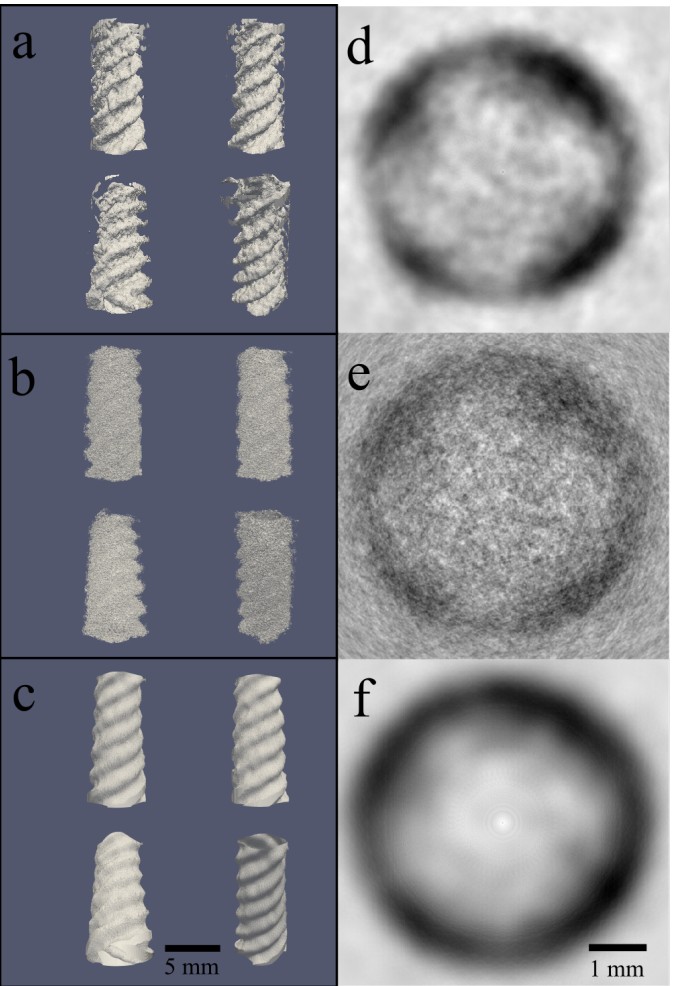

**Figure 2: Comparison of reconstructed $^1$H-NMR spin noise tomograms of the phantom immersed in $H_2O/D_2O$ (9:1) recorded at 700 MHz obtained with three different processing procedures from the same raw data. The acquired signal originates from the water protons around the solid phantom, which is not observable due to the extremely broad lines of the polymer. Total noise data**

**recording time was 40 h. (a-c): Four different views for each processing procedure visualized as iso-surfaces representing the boundary between liquid and solid. (d-f): Comparison of density plots of (X-Y) 2D cross-section extracted from the centre (along the Z direction) of the corresponding 3D image of the $^1$H noise magnitude signal (white representing zero amplitude and black the maximum magnitude). Panels (d), (e) and (f) display 2D cross-section of the 3D images shown in in (a), (b) and (c), respectively. (a), (d): Images obtained by our new iterative reconstruction processing using combinations of different time-domain sliding window**

**sizes (as described in the main text). (b), (e): Images processed with the standard projection reconstruction algorithm (Andersen and Kak, 1984), using a time domain sliding window size of 1024 data points (corresponding to 103 ms), resulting in high resolution but low signal-to-noise ratio (contrast). (c), (f): Images processed with standard projection reconstruction using a time domain sliding window size of 128 data points (corresponding to 13 ms), resulting in low resolution and higher signal-to-noise ratio (contrast).**

This resolution is less than the maximum obtainable one based on the experimentally determined water proton transverse

relaxation time (380 ms). But increasing the window size beyond 1024 data points reduces the SNR (signal-to-noise ratio) too



much. The same raw data processed with a small window size (128 data points) results in a correspondingly higher SNR but achieves an overly smoothed representation of the phantom a shown in Fig. 2c and f. The window length of the smaller blocks was chosen empirically by halving the length of the longer windows until the SNR was acceptable, while still being able to resolve courser details in the image.

In Fig. 2d-f images of cross sections of the phantom obtained by the different processing schemes are compared, demonstrating the flexibility of adjusting the contrast/resolution trade-off *a posteriori* from the same raw data. The high-resolution image (Fig. 2c and f) obtained by the conventional SART method might accurately represent the phantom, but the low SNR makes it difficult to draw a clear separation line between the phantom and the surrounding water in the cross-section in Fig. 2f.
The low-resolution image in Fig. 2b and e, also obtained by SART, shows this boundary more clearly, but the resolution is too

low to make out the correct shape of the phantom in the cross-section in Fig. 2e. Only the image calculated with the iterative reconstruction method (Fig. 2a and d) shows a phantom with clear boundaries and a non-circular shape in the cross-section in Fig. 2d, where the four "pockets" of water formed by the phantom can be resolved.
In Sect. 3 we describe the new processing procedure yielding the images in Fig. 2a and d, which, to our knowledge, is the most efficient way to obtain 3D images from spin noise data, currently. Most notably the continuous nature of spin noise time

domain data allows one to decide the resolution vs. contrast trade-off by reprocessing the raw data.
For completeness, we discuss the concept of indirect detection of the spatial dimension in Fourier imaging with spin noise. Even though spin noise has random phase, one can devise a way to encode spatial information in the indirect dimension similar to the way used for 2D spin-noise-detected spectroscopy (Chandra et al., 2013) (Ginthör et al., 2018). This would require a location-encoding gradient sandwiched between two acquisition blocks and incremented in the usual way to yield the indirect

k-space dimension. This phase-encoding gradient modulates the relative phase of the signal, which could be resolved by cross-correlation of two subsequent acquisition blocks. However, this theoretical scheme suffers from excessive relaxation losses occurring during this gradient and the two acquisition periods involved. We have so far not succeeded in obtaining sufficient image contrast using this scheme. Notably, with this indirect time domain (k-space) approach one cannot take advantage of the sliding window processing and *a posteriori* optimisation as this acquisition scheme only affords discrete short data blocks.

**3. Materials and Methods**

The experiments were carried out on a high-resolution NMR spectrometer equipped with a Bruker Avance III console connected to a Bruker Ascend 700 MHz magnet and a TCI cryoprobe (manufactured in 2011). The phantom is a 3D-printed helix made of PLA (polylactic acid) fitting tightly inside a standard 5 mm NMR tube (Wilmad 535-PP). The tube is filled with a mixture of $H_2O{:}D_2O$ (9:1) and the filling height made equal to the height of the phantom (50 mm). The rf probe is tuned to

the SNTO (spin noise tuning optimum) (Marion and Desvaux, 2008) (Nausner et al., 2009) (Pöschko et al., 2014) and 3D shimming (using Bruker's TopShim) is performed on a sample (henceforth referred to as "shim sample") which is prepared in the same type of 5 mm tube (Wilmad 535-PP) as the phantom but filled with the same mixture of $H_2O{:}D_2O$ (9:1) to the same filling height without the phantom. A one-dimensional (1D) [1]H spin noise spectrum is acquired to verify the correctness of the



setup and the shim. For this imaging experiment the magnetic field gradients generated by the X, Y, and Z field correction

(shim) coils are used as the imaging gradients, after calibration for the purpose of imaging. The gradient amplitude is measured

by the broadening of the peak in a spectrum of the test sample. For each projection direction the magnitude of the applied

magnetic field gradient must be the same, independent of the direction set according to the $\varphi$ and $\theta$ values. (The definitions of

the coordinate system and angles are given in the Supplementary Information.) This is achieved by calculating the individual

shim values via the trigonometric laws. A few compound gradient settings (involving multiple individual gradients) are

checked to verify orthogonality of the Cartesian field gradient components, that are used to calculate the gradient directions.

After setup and calibration, the actual phantom is inserted into the magnet without further shimming. A script controls the

setting of the gradients via the digital-to-analogue converters of the shim system and initiates the acquisition. For each angle

$\varphi$ and $\theta$ 30 projections were recorded, yielding 900 projections in total at a gradient amplitude of 78 mT/m. The values are

sampled uniformly per angle in the range of 0 to 180°. The projections were recorded with the following settings: For each

projection angle two noise blocks were recorded (time domain points: 1024 k, spectral width: 5 kHz, spectral centre: 4.670

ppm). The individual projections are recorded like standard 1D spin-noise-detected spectra (Müller and Jerschow, 2006)

(Nausner et al., 2009). The acquisition sequence is illustrated and compared to conventional pulsed NMR in Fig. 3.

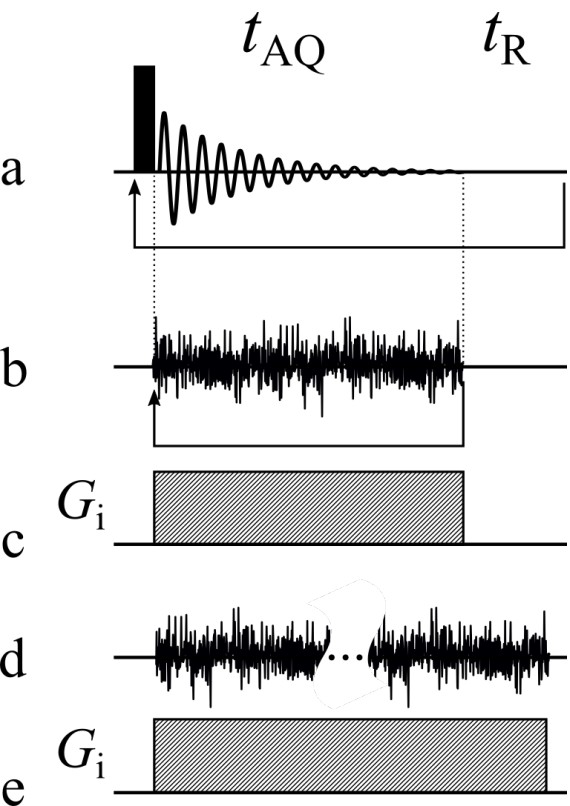

**Figure 3: Basic acquisition sequences. (a) Acquisition sequence of the simplest 1D pulsed experiment. (b) The corresponding spin-**
**noise-detected NMR acquisition sequence. The arrows in (a) and (b) represent the loops for recording all the scans. (c) The**
**frequency-encoding gradient for (a) and (b) if used as an imaging experiment. (d) Same as (b), but all data blocks are recorded**



**continuously in one long acquisition period. (e) The frequency-encoding gradient for (d) if used as an imaging experiment. $t_{AQ}$ and $t_R$ denote the acquisition time and recycle delay, respectively.**

In a first approximation, noise acquisition can be modelled as analogous to the simplest conventional 1D NMR experiment (Fig. 3a), with a very short random phase excitation pulse. The direction of the projection is determined by the magnetic field gradient active during acquisition. Gradients for different projection directions are set to the same magnitude and thus only differ in their direction. The angles are laid out in the following way: $N$ angles for $\varphi$ (angle between X-axis and XY-projection of direction vector) are chosen uniformly. For each angle $\varphi$ $M$ angles for $\theta$ (angle between Z-axis and direction vector) are

chosen uniformly. A reference coordinate system can be found in the Supplementary Information. The $\theta$ angles have the same set of values for all $\varphi$ angles.

    Due to the non-deterministic nature of the spin noise phase, it is not possible to accumulate the raw phase sensitive data directly in the time-domain (as it is usually done), as this would lead to signal cancellation. Instead, all noise blocks (the spin noise equivalent of "FIDs") are stored and Fourier-transformed individually. After calculating the power or magnitude spectra their

addition yields the final projection (McCoy and Ernst, 1989) (Müller and Jerschow, 2006) (Nausner et al., 2009). Relaxation not being an issue any recycling delay can be omitted. That allows one to record all blocks for one gradient in a single very long acquisition block, as indicated in Fig. 3d. During processing the data is split into adjustable smaller blocks which can be processed as described above for individually acquired ones. The acquisition of a single large noise block also allows one to take advantage of "sliding window processing", which was introduced by Desvaux et al. (Desvaux et al., 2009). By slicing a

long acquisition block recorded into overlapping sub-blocks one gains the option to compromise *a posteriori* (i.e. after acquisition of the raw data) between resolution and sensitivity, by adjusting the size of the smaller sub blocks. The optimum overlap of the blocks or "selection windows" has been shown to be approximately 1/7 of the windows' size (Desvaux et al., 2009). Due to this overlap seven times more blocks are used in the summation process, which improves the SNR by a factor of about √2, owed to the different accumulation behaviour of correlated (spin) noise and uncorrelated (instrument and circuit)

noise. After the sliding window splitting, the individual blocks are processed in the same way as individually acquired ones. The processing schemes from time-domain noise data to the final projection with and without the sliding window processing are compared in Fig. 4.

    The processing was achieved by a custom Python script (available in the Supplementary Material), using the library numpy

(Oliphant, 2006) for general numerical calculations and for the FFT routine as well as scikit-image (Walt et al., 2014) for implementing the 2D SART reconstruction algorithm (see below). The iterative reconstruction method was used with two different resolutions. The lower resolution images were processed with a sliding window length of 128 and an overlap of 110 data points, the higher resolution ones with a length of 1024 and an overlap of 878 data points. The resulting three-dimensional (3D) images were visualized with ParaView (Ahrens et al., 2005).



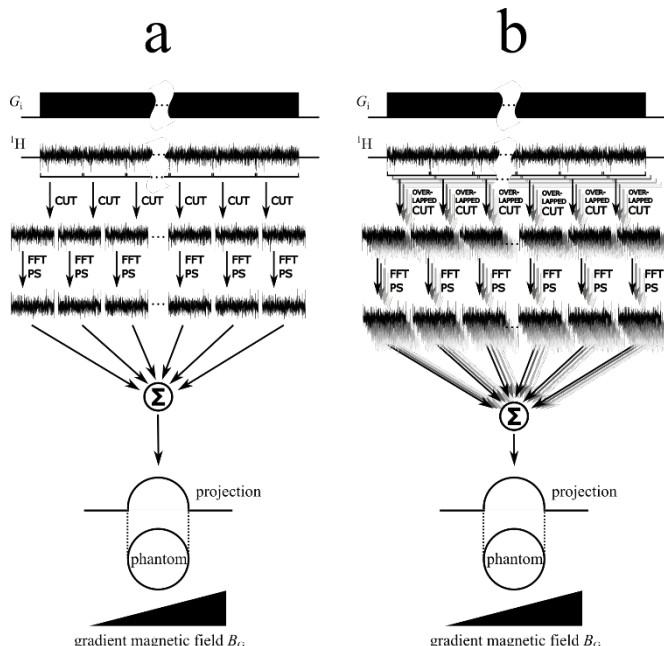

**Figure 4: Comparison of sliding window processing schemes for spin noise data without (a) and with (b) overlapping windows. The acquisition sequence (including the frequency-encoding gradient magnetic field $B_G$) for recording one long noise block is shown on top. In the second step this block is cut up into multiple smaller noise blocks. After the fast Fourier-transformation (FFT) and calculating the power spectra (PS), the sum of all individual spectra yields the final projection. In (b) the long noise block is cut up into overlapping sub-blocks, yielding a higher number of smaller noise blocks.**

We found the simultaneous algebraic reconstruction technique (SART) introduced by Andersen and Kak (Andersen and Kak, 1984) to be most suitable as an alternative to the inverse Radon transform for the spin noise imaging data. This algorithm sets up a set of linear equations (see Eq. (1)) that describe the dependency of the projected values on the distribution function D($r$) of the original sample (Andersen and Kak, 1984).

$$\wp_{\beta,i} = \int_A D(r) \, dr \tag{1}$$

Integration occurs over the area $A$, $\wp_{\beta,i}$ denotes the data point $i$ in the projection $\wp$ at the angle $\beta$.

The vector $\boldsymbol{r}$ encodes the position of a point inside the sample. In our case, $A$ is an area along the projection direction through the sample with the width of one data point. No exact solution of this system is possible because it is usually underdetermined (owed to a finite number of projection values $\wp$ compared to a continuous distribution function). Therefore D($r$), the solution of this system of linear equations is approximated by an appropriate method, e.g. the Kaczmarz method (Kaczmarz, 1937). For two dimensions the equations can be generated in the following way: Given $N$ data points for each projection, an image matrix $\mathfrak{I}$ of $N$ x $N$ pixels is created, representing the reconstructed image. The image matrix $\mathfrak{I}$ is a discrete approximation of the continuous distribution function D($r$). There are multiple ways to formulate the forward projection process, but here a bilinear model is chosen in Eq. 2, as it is the basis for the SART reconstruction method (Andersen and Kak, 1984).





$$\wp_{\beta,i} = \sum_{j=1}^{S} \sum_{k=1}^{4} w_{\mathcal{L}_j,k} \mathfrak{I}_{\mathcal{L}_j,k} \tag{2}$$

$\wp_{\beta,i}$ denotes the data point $i$ in the projection $\wp$ at the angle $\beta$. The image matrix $\mathfrak{I}$ is sampled $j$ times along the projection
direction for each point $i$ in the projection $\wp_\beta$.

The resulting value for the sampling location is the weighted sum of the four closest pixels (index $k$) in the image matrix,
where w denotes the weighting factor for the individual pixel. This also shows the underdetermination of the system. There
are $N^2$ variables (pixels in image matrix) but only $x * N$ equations ($x$ represents the number of projection angles). The
projections can be conceptually arranged around the matrix $\mathfrak{I}$ at their respective projection angles. The reconstruction works
in the following way: From each projection $\wp_\beta$ (corresponding to angle $\beta$) an updated image is calculated according to Eq. (3)
(Andersen and Kak, 1984).

$$\mathfrak{I}_{z+1} = \mathfrak{I}_z + u(\mathfrak{I}_z, \wp_\beta) * r \tag{3}$$

Here u denotes the update function, $\wp_\beta$ the projection at angle $\beta$ and the relaxation factor $r$. $z$ indicates the evolution of the
image matrix after an update has been applied. All projections $\wp$ are processed sequentially in an order that maximizes the
difference in angles between two successive projections. The reconstruction is complete when all the projections $\wp$ have been
considered. The goal is to approximate $D(r)$ with $\mathfrak{I}$. The update function u performs three steps for each projection pixel $\wp_{\beta,i}$.
First, from each projection pixel $\wp_{\beta,i}$ a ray is cast into $\mathfrak{I}$, along the projection direction. $S$ intermediate values of the matrix $\mathfrak{I}$
are taken on that ray at equidistant locations $\mathcal{L}_j$. At each location the values of the four closest matrix $\mathfrak{I}$ pixels, weighted by
their distance to the sampling point are summed to give the value assigned to the sampling location $\mathcal{L}_j$. In Fig. 5 this process is
illustrated.

The sum of all sampling points along the projection ray yields a new value for each projection pixel $\wp_{\beta,i}$, named $\wp_{calc\,\beta,i}$
according to Eq. (4).

$$\wp_{calc\,\beta,i} = \sum_{j=1}^{S} \sum_{k=1}^{4} w_{\mathcal{L}_j,k} \mathfrak{I}_{\mathcal{L}_j,k} \tag{4}$$

The index variable $j$ spans all sampling locations $\mathcal{L}_j$ along the ray, $k$ is used to indicate the four surrounding pixels of a single
sampling location $\mathcal{L}_j$; w is the weight used to calculate the contribution of the neighbouring pixel in question. This is repeated
for all projection pixels $\wp_{\beta,i}$ of a given projection $\wp_\beta$. The difference in value of the actual projection pixel $\wp_{\beta,i}$ and the
calculated projection pixel $\wp_{calc\,\beta,i}$ is given in Eq. (5).

$$\wp_{diff\,\beta,i} = \wp_{x,i} - \wp_{calc\,\beta,i} \tag{5}$$



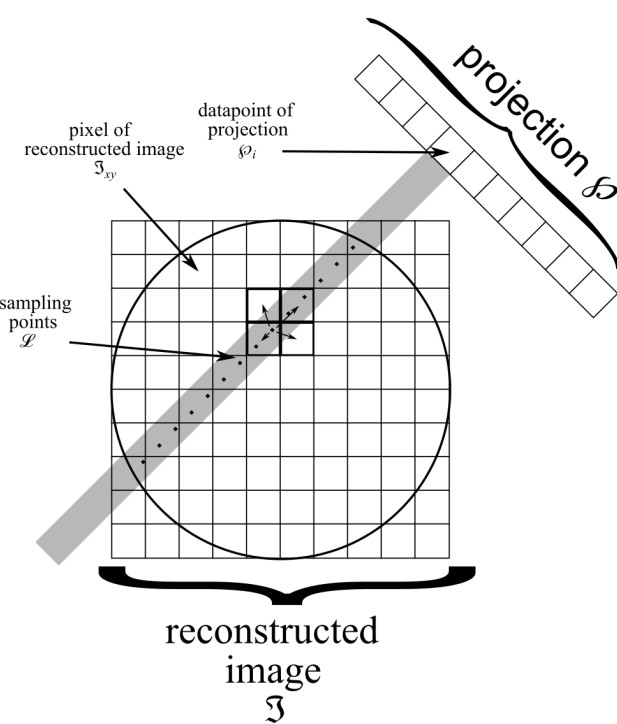


**Figure 5: Scheme of our implementation of the SART algorithm (Andersen and Kak, 1984). A ray is cast from every data point of the projection $\wp$ into the image matrix $\mathfrak{I}$. Along this ray $S$ values of the matrix elements (pixels) are sampled from $\mathfrak{I}$ at locations $\mathcal{L}_j$ ($j = 1..S$). Each point $\mathcal{L}_j$ is computed as the weighted sum of the four closest points (pixels) of $\mathfrak{I}$.**

This difference is then distributed back to the individual sampling locations and in turn to their surrounding pixels,  weighted

by the respective distance from the sampling location to the neighbouring pixel, in a new previously empty image matrix $\mathfrak{I}_{update}$

as seen in Eq. (6).

$$\mathfrak{I}_{update\ \mathcal{L}_j,k} = \wp_{diff\ \beta,i} * w_{\mathcal{L}_j,k} \qquad\qquad (6)$$


This matrix $\mathfrak{I}_{update}$ is the result of the update function u and used to update the image matrix in Eq. (3).

The algorithm can be initiated with a first guess of the image matrix $\mathfrak{I}_z$=0. Starting with a reasonable guess as the initial matrix

instead of a zero matrix improves the reconstruction, provided the guess is not too far off, in our case we use a low-resolution

image.

The reconstruction is iterated a pre-determined number of times (in our case 2) with the same projections, but always using

the result from the previous step as the next initial guess. In each step the weight of narrow contributions increases (e.g. better-

defined edges or increased contrast), while at the same time the noise level is rising. The number of iterations depends on the



quality of the original projections and is, for actual data, best determined by visual inspection of the image improvement during the iteration process. The relaxation factor $r$ in Eq. (3) was set to 0.05 for all reconstructions.

The projections are grouped by the values of the angle $\varphi$, yielding $N$ groups with $M$ projections each. For each of these groups a SART image reconstruction (Andersen and Kak, 1984) is performed, yielding $N$ 2D images. The rows (with respect to the Z-axis) with the same index from all the images are grouped again. Every first row in a group, every second in a group and so on. Again, for each of these groups an image reconstruction is performed. This results in the final 3D image of the phantom. Figure 6 illustrates this procedure.


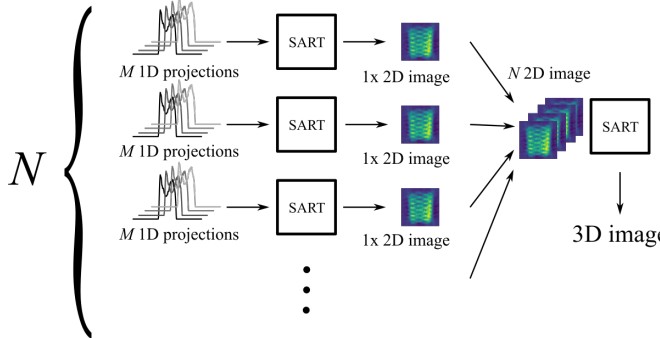

**Figure 6: 3D SART reconstruction process. The 3D SART reconstruction algorithm is realized as a two-step process via two successive two-dimensional SART reconstructions. $M*N$ 1D projections are grouped by same $\varphi$ angle ($N$ groups in total). Each of the $N$ groups contains $M$ projections with different angles $\theta$. The first SART reconstruction yields $N$ 2D images. Those images**
**represent two-dimensional projections of the sample, all perpendicular to the XY plane. The individual angle between each 2D image and the x-axis corresponds to the $\varphi$ angle of the projections the image was created from. Every row (along the Z axis) of every 2D image then represents a 1D projection of a XY slice of the sample (at the same height as the row) around the Z axis. The final 3D image can be reconstructed by using a 2D SART reconstruction on each of those rows individually.**

In conjunction with the SART image reconstruction the flexibility of large noise block acquisition and the sliding window processing turns out to be particularly advantageous. Shorter windows yield a higher number of blocks with fewer data points and hence lower resolution. The summation over a larger number of noise blocks, however, improves the SNR of the resulting projection. Larger windows have the opposite effect and improve the resolution while losing SNR. This fact is exemplified in Fig. 7. The initial image reconstruction (via SART, see above) is carried out with a zero seed matrix using projections computed
from short sliding windows (e.g. 128 complex data points).





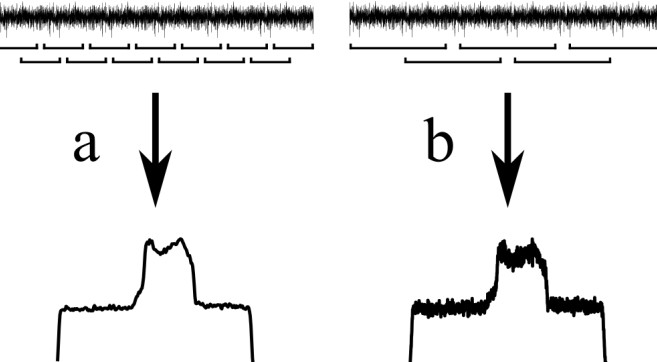

**Figure 7: Qualitative comparison of resolution and signal-to-noise ratio for different sliding window sizes using a 1D image of the phantom. (a) shows the result of the sliding window algorithm with a small windows size, yielding a low-resolution image with a high signal-to-noise ratio. (b) shows the result of the same process with a longer sliding window, resulting in a spectrum with higher resolution but lower signal-to-noise ratio as compared to (a).**

The reconstructed image from these low-resolution projections then serves as the starting image matrix for the second reconstruction using projections calculated with a longer sliding window. A schematic overview of the procedure can be seen in Fig. 8.

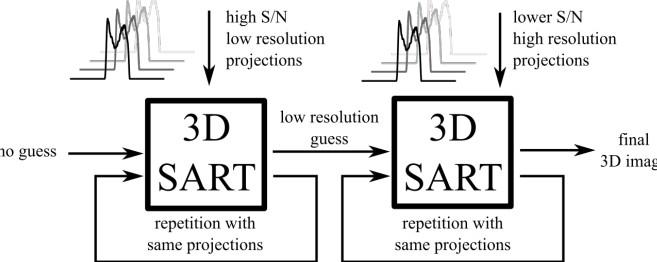

**Figure 8: Iterative 3D SART reconstruction. The first 3D SART reconstruction uses no initial guess and reconstructs a low-resolution image with a high signal-to-noise ratio. This intermediate result is used as the initial guess for the second 3D SART reconstruction, yielding the final reconstructed image. Each 3D SART reconstruction is repeated multiple times with the same projections.**

For additional clarity, we summarize the iterative reconstruction process: The recorded noise blocks corresponding to different projection angles are processed with different increasing sliding window sizes. This procedure yields multiple sets of projections, that differ in their resolution and SNR. Then a first 2D SART image reconstruction step is done separately for each set of 1D projections corresponding to a particular angle $\varphi$. The first reconstruction uses no initial guess (i.e. a zero image matrix) and is reconstructed using the set of projections with the lowest resolution and correspondingly highest SNR. The obtained 2D image is used as initial guess for another reconstruction, which is improved by the next set of projections with higher resolution but lower SNR. Continuing this series, the obtained 2D images now form a sequence ranging from low resolution, high SNR images to the inverse high resolution, low SNR case. In order to use the lower resolution images in the

higher resolution reconstructions, the image matrices need to be interpolated to match the required number of data points. The
sliding windows sizes are doubled on each incrementation such that the resolution always increases by a factor of 2. This
procedure is used analogously in the second reconstruction step to obtain the final 3D image.

## 4. Conclusion

We have improved spin-noise-detected NMR imaging and extended it to three dimensions from the original 2D technique
published in 2006 (Müller and Jerschow, 2006). Exploiting using tuning optimization and the unique properties of spin noise,
in particular that it does not decay and has no defined starting point in time, it is possible to use a quasi-continuous acquisition
technique yielding raw noise data of arbitrary duration while applying constant magnitude magnetic field gradients in different
directions. These data can be processed in a unique way using a dynamically adapted sliding window to define fractionally
overlapping data blocks, which Fourier transformed and are co-added after power spectrum computation.

Based on these fundamental steps, we have introduced a new kind of SART-based iterative image reconstruction technique,
which yields 3D images that are superior in visual quality improving SNR and resolution at the same time without introduction
of artefacts. In this type of spin noise imaging the well-known trade-off between SNR (or image contrast) and resolution can
be adjusted *a posteriori* during processing of the same original data by iterative image reconstruction, which is not applicable
in conventional rf pulse dependent MRI.

### Team list

Stephan J. Ginthör, Judith Schlagnitweit, Matthias Bechmann, Norbert Müller

### Author contribution

N.M. planned and supervised the research; S.J.G. performed all experiments, programmed the processing algorithms and
processed the data; J.S. devised the time domain spin noise imaging experiment and contributed to the manuscript; M.B. and
S.J.G. programmed the spectrometer; N.M. and S.J.G wrote the manuscript.

### Competing interest

The authors declare no competing interest.



**Acknowledgments**

This work has been supported by the European Union through the ERDF INTERREG IV (RU2-EU-124/100--2010) program
(ETC Austria-Czech Republic 2007-2013, project M00146, "RERI-uasb" for N.M.).

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
