# Peer review of "Nuclear spin noise tomography in three dimensions"

_Magnetic Resonance, 2020_

## Short Comment (SC1) · 5 Jun 2020

The fact that this remarkable paper has not attracted any comments so far confirms my impression that the MRI community has become very conservative.

---

## Referee Comment (RC1) · Anonymous Referee #1 · 13 Jun 2020

This manuscript demonstrated 3D spin-noise imaging. While fundamentally, the work does not go beyond previously-demonstrated 2D spin-noise imaging, since the earlier demonstration there has been a lot of progress in understanding the spin-noise phenomenon, developing optimized processing strategies, describing spin-noise lineshapes, and optimal tuning conditions, all information that has gone into performing the 3D imaging work presented here. Therefore, I view this work as important in highlighting and summarizing critical aspects of spin-noise detection, and point to potential future applications. Furthermore, I find the image reconstruction in Fig. 2 particularly striking, especially given that pure spin noise is used to acquire the data. Given the weakness of the spin-noise signal, the work is also a demonstration of sensitivity limits of today's NMR spectroscopic equipment. The use and evaluation of the SART

technique is interesting, especially since it is not used much in image reconstruction. One (very) minor comment is that the conclusion says "spin-noise does not decay". This may be a bit misleading, since, spin-noise rather than decaying loses memory at a time scale of T2. Another minor comment would be that it may be useful to state whether SART could be implemented in a 3D fashion rather than in the pseudo-3D approach used here. Overall, I find this to be great and nicely executed work, which is a wonderful addition to the journal.

---

## Referee Comment (RC2) · Anonymous Referee #2 · 21 Jun 2020

Authors properly applied the spin noise phenomena in 3D tomography of specific object immersed in the solvent. The remarkable beauty of spin noise lays in naturally performing NMR experiments without disturbance to spins by gently listening what spins can reveal about themselves and not using RF pulses which they normally act as brute force. Spin noise coherence does not need to be created by pulses as such already exists being created by the nature of the spin fluctuations statistic. Presented manuscript is continuation of previous work done on the spin-noise-detected NMR imaging in two-dimensions published in 2006. Since then authors, made an excellent progress in researching the spin noise phenomena in several aspects not limited to imaging as well as in an optimization of associated hardware and software. This allowed to demonstrate much better visual quality images of phantom thanks to introducing of a new

kind of SART-based iterative image reconstruction technique.

I would like to suggest a few minor comments for consideration.

Since sensitivity of the spin noise is closely related to the magnitude of radiation damping, I would like to suggest some comment on this issue. It would be helpful to know the radiation damping constant of the H2O/D2O system and compare it to the T2 spin-spin relaxation time. Since radiation damping is involved in providing a coupling between spin system and RF coil this has a very significant impact on practicability, efficiency and successful application of spin noise.

160 pg.7 "In a first approximation, noise acquisition can be modeled as analogous to the simplest conventional 1DNMR experiment 160(Fig. 3a), with a very short random phase excitation pulse." This is not accurate enough statement. Simplification could be misleading. By no any means one can perform NMR experiment with comparable number of excitation pulses to the number of spins. Therefore the statistic in both cases will be very different. Each spin posses its own phase and typically contributes to the magnetization and overall statistic gives M~square root[N]. Simplest conventional NMR with very short random pulses will not yield such relation.

160 pg.7 "For each angle $\varphi$M angles for $\theta$", I would suggest considering a different character for M as in NMR this symbol is generally reserved for the magnetization.

165. pg.7 "Due to the non-deterministic nature of the spin noise phase, it is not possible to accumulate the raw phase sensitive data directly in the time-domain (as it is usually done), as this would lead to signal cancellation." The statement "Non-deterministic" needs future explanation. Is non-deterministic nature because of uncertainty principle or simple due to the phase time dependence and lack of possibility to acquire enough signal to be observed at the unique phase value at the acquisition time adequate to the linewidth? In physics, the statement "non-deterministic nature" rises often a lot of ambiguity.

170 pg.7 "Relaxation not being an issue any recycling delay can be omitted" This is too generic statement especially when later in the manuscript the relaxation factor in eq.(3) is used with different meaning. I would suggest being more specific and add spin-lattice relaxation. On the other hand spin-spin relaxation is still relevant and important.

Significant part of the manuscript is devoted to SART-based iterative image reconstruction technique. However, this does not have explicit reflection in the title of the article. I would suggest considering including in the title the statement "SART" so this could better reflect the scope of the work as well as improve search-ability of the article.

315 pg.13 "unique properties of spin noise, in particular that it does not decay and has no defined starting point in time". "Spin noise does not decay", is not an accurate statement. Spin noise originates by spin fluctuations which they exist all of the time. By the property of such fluctuations they will never disappear and at the same time they will decay. Autocorrelation of fluctuations exhibits an exponential behavior which mirrors the free induction decay. On the other hand the linewidth of spin noise spectrum is related to T2 relaxation that is associated with FID which always involves loosing the phase coherence and magnetization decay. If spin noise does not decay, this naturally requires T2 relaxation time being infinitely long and linewidth should approach 0.0 Hz which is beyond objective reality.

The manuscript is revealing a significant progress in 3D spin noise tomography and I am recommending it for publishing after considering these minor issues.

---

## Author Comment (AC1) · 9 Jul 2020

As formulae and special characters are included the pdf version of this reply is appended as a supplement.

Reply to the comments of anonymous Referee #1:

We are very grateful for and flattered by the positive comments. The importance of the processing algorithm has been stressed by a change of the articles title which was suggested by reviewer #2.

•"spin-noise does not decay" Concerning the phrase "spin-noise does not decay" we agree that this is misleading and have recast the formulation to be more physically precise in the Conclusion Section. The issue has also been raised by reviewer #2.

[Figure]

The unique properties of spin noise, in particular its average power being constant, while phase memory is lost with the usual time constant T2*, and the absence of a defined starting point in time together with spin noise tuning optimization (Nausner 2009, Pöschko 2014) make it possible to use a quasi-continuous acquisition technique. Thus, one can obtain raw noise data of arbitrary duration while applying constant magnitude magnetic field gradients in different directions..

• The use and evaluation of the SART technique is interesting, especially since it is not used much in image reconstruction

With respect to the 3D SART method, we are aware of the existence of commercial 3D implementations for MRI, which are not easily accessible within the academic world. In the revised manuscript we will append at the end of Section 3:

"We note that the SART algorithm can also applied to all three dimensions in a single step without intermediate 2D data (Mueller et al. 1999), but this has not been done here due to higher computational demands and the easy availability of well-tested open-source 2D SART variants."

Reply to the comments of anonymous Referee #2:

We'd like to thank for the positive comments and very constructive criticism and suggestions. Below, we are addressing the specific comments in order of occurrence:

• Since sensitivity of the spin noise is closely related to the magnitude of radiation damping, I would like to suggest some comment on this issue. It would be helpful to know the radiation damping constant of the H2O/D2O system and compare it to the T2 spin-spin relaxation time. Since radiation damping is involved in providing a coupling between spin system and RF coil this has a very significant impact on practicability, efficiency and successful application of spin noise.

To address this issue, we made additions in two places in the revised manuscript:

We added the definition of the radiation damping rate constant as Eq. 1 (p.3 l.77)

together with a note to its importance for the sensitivity.

"While the sensitivity of spin noise acquisition depends in a highly non-linear way on the radiation damping rate $\lambda\_rd^0$ (McCoy and Ernst, 1989), (Nausner et al., 2009), (Pöschko et al., 2017) in this imaging regime it is proportional to the radiation damping rate at equilibrium.

$$\lambda\_rd^0 = \mu\_0/2\ \eta Q\gamma M\_0 \quad (1)$$

Here, the only variables which depend on the instrumental setup are the filling factor $\eta$, and the probe quality factor Q, while the gyromagnetic ratio $\gamma$ and $\mu0$, the permeability of vacuum are immutable. Note that apart from maximizing the coupling between the spins and the rf-circuit by increasing $\eta$ and Q the SNR (signal-to-noise ratio) of spin noise detected experiments can be improved by reducing the noise from all other sources.

To provide some comparable values for different setups we inserted in the Materials an Methods Section:

"The radiation damping rate of this phantom in the probe used was determined as $\lambda\_rd^0 = 611.4$ rad/s under the conditions of the imaging experiment while the transverse relaxation rate in the absence of radiation damping was $1/T2 = 2.632$ rad/s."

• 160 pg.7 "In a first approximation, noise acquisition can be modeled as analogous to the simplest conventional 1DNMR experiment 160(Fig. 3a), with a very short random phase excitation pulse." This is not accurate enough statement. Simplification could be misleading. By no any means one can perform NMR experiment with comparable number of excitation pulses to the number of spins. Therefore the statistic in both cases will be very different. Each spin posses its own phase and typically contributes to the magnetization and overall statistic gives MâĹijsquare root[N]. Simplest conventional NMR with very short random pulses will not yield such relation. "

The analogy we have used is admittedly crude and not even a "first approximation"

which is not even required in that context. We actually are using this analogy mostly internally when testing spin noise-based coherence transfer experiments by inserting a short pulse for quick testing. But that is not the case in the experiments we report here. So, we will remove this passage and just refer to the mode of generating images from field gradients during acquisition. (Now p.8 ll 195ff):

"As in a conventional single pulse NMR experiment (Fig. 3a), the direction of the projection is determined by the magnetic field gradient active during acquisition."

• "160 p.7 "For each angle $\varphi$M angles for $\theta$", I would suggest considering a different character for M as in NMR this symbol is generally reserved for the magnetization. "

Agreed, the symbolic name space is getting crowded. We now use U and V instead of N and M. throughout the MS. Figure 6 has been updated to U and V as well, because it referenced M and N.

• l165. pg.7 "Due to the non-deterministic nature of the spin noise phase, it is not possible to accumulate the raw phase sensitive data directly in the time-domain (as it is usually done), as this would lead to signal cancellation." The statement "Non-deterministic" needs future explanation. Is non-deterministic nature because of uncertainty principle or simple due to the phase time dependence and lack of possibility to acquire enough signal to be observed at the unique phase value at the acquisition time adequate to the linewidth? In physics, the statement "non-deterministic nature" rises often a lot of ambiguity.

For sure we did not mean non-deterministic nature because of uncertainty principle. I'd prefer to leave the philosophical considerations, which would extend into a prolongation of the long-standing discussion of spin noise being spontaneous or induced. So, we write write in the revision, leaving the source of randomness open (Now: p.8 l.201):

"Due to the random phase of spin noise, direct accumulation of raw phase sensitive data in the time-domain (as it is usually done for pulsed experiments) leads to signal

cancellation"

• 170 pg.7 "Relaxation not being an issue any recycling delay can be omitted" This is too generic statement especially when later in the manuscript the relaxation factor in eq.(3) is used with different meaning. I would suggest being more specific and add spin-lattice relaxation. On the other hand spin-spin relaxation is still relevant and important.

"Relaxation" is indeed to broad a term here. In the revision we will change that part to:

"Longitudinal relaxation not being an issue in the absence of rf-pulses any recycling delay can be omitted."(now p.8 l 205)

In the context of the SART algorithm we will clarify (p.10 l.237):

"the image reconstruction relaxation factor r (with no relation to spin relaxation)." (now l.285 p.10)

• Significant part of the manuscript is devoted to SART-based iterative image reconstruc- tion technique. However, this does not have explicit reflection in the title of the article. I would suggest considering including in the title the statement "SART" so this could better reflect the scope of the work as well as improve search-ability of the article.

We will change the MS title to: "Nuclear spin noise tomography in three dimensions with iterative SART processing", provided this is OK for the Editors.

• l315 pg.13 "unique properties of spin noise, in particular that it does not decay and has no defined starting point in time". "Spin noise does not decay", is not an accurate statement. Spin noise originates by spin fluctuations which they exist all of the time. By the property of such fluctuations they will never disappear and at the same time they will decay. Autocorrelation of fluctuations exhibits an exponential behavior which mirrors the free induction decay. On the other hand the linewidth of spin noise spectrum is related to T2 relaxation that is associated with FID which always involves loosing the
phase coherence and magnetization decay. If spin noise does not decay, this naturally requires T2 relaxation time being infinitely long and linewidth should approach 0.0 Hz which is beyond objective reality.

We agree with the reviewer's comments and will change that part of the Conclusion Section to (now: p.13 l.398):

The unique properties of spin noise, in particular its average power being constant, while phase memory is lost with the usual time constant T2*, and the absence of a defined starting point in time together with spin noise tuning optimization (Nausner 2009, Pöschko 2014) make it possible to use a quasi-continuous acquisition technique. Thus, one can obtain raw noise data of arbitrary duration while applying constant magnitude magnetic field gradients in different directions.

Please also note the supplement to this comment:
https://mr.copernicus.org/preprints/mr-2020-12/mr-2020-12-AC1-supplement.pdf

―――――――――――――――――――――――

---

## Editor Comment (EC1) · Matthias Ernst (Editor) · 10 Jul 2020

Dear Norbert,

please submit a revised manuscript with the changes outlined in your reply to the review reports. If the revision cover all these points, I will accept the manuscript for publication in Magnetic Resonance.

Best regards,

Matthias Ernst